# Work intensity and fat mass percentage are associated with asymptomatic morphometric vertebral fractures in knee osteoarthritis patients: A cross-sectional study

Izzatul Nadiah Zolkiply[1], Kah Keng Wong[2], Hakimah Mohammad Sallehudin[1], Mohammad Zulkarnain Bidin[1], Fahrudin Che Hamzah[3], Norafida Bahari[4], Wan Syamimee Wan Ghazali[1,5]*

1 Department of Medicine, Faculty of Medicine and Health Sciences, Universiti Putra Malaysia, Serdang, Selangor, Malaysia, 2 Department of Immunology, School of Medical Sciences, Universiti Sains Malaysia, Kubang Kerian, Kelantan, Malaysia, 3 Department of Orthopedic, Faculty of Medicine and Health Sciences, Universiti Putra Malaysia, Serdang, Selangor, Malaysia, 4 Department of Radiology, Faculty of Medicine and Health Sciences, Universiti Putra Malaysia, Serdang, Selangor, Malaysia, 5 Department of Internal Medicine, School of Medical Sciences, Universiti Sains Malaysia, Kubang Kerian, Kelantan, Malaysia

* syamimee@usm.my

**Data Availability Statement:** All relevant data are within the paper.

## Abstract

Knee osteoarthritis (OA) is a common condition with a prevalence of 365 million individuals globally, and it is an independent risk factor for falls and fractures, notably asymptomatic morphometric vertebral fractures (AMVF). The high prevalence of knee OA, the severity of AMVF, and their combined impacts on quality of life underscore the need for early detection, appropriate treatment and management. To address this, our cross-sectional study aims to identify potential predictive factors associated with AMVF in knee OA patients. Our cohort consisted of 76 patients diagnosed with knee OA, predominantly female (84.2%), of Malay ethnicity (84.2%), and obese (55.3%). In univariable analysis, significant association was found between occupation (moderate or heavy work) and AMVF ($p<0.001$). Diabetes mellitus comorbidity ($p = 0.016$) and fat mass percentage ($p = 0.027$) also demonstrated a significant association with AMVF in knee OA patients. Multivariable logistic regression analysis revealed that an increase in fat mass percentage resulted in decreased AMVF incidence (HR: 0.83, 95% CI: 0.70–0.97; $p = 0.018$), while occupation (moderate or heavy work) remained a highly significant predictor (HR: 57.76, 95% CI: 4.23–788.57; $p = 0.002$). These findings support the potential importance of considering occupational activities and body fat composition in managing AMVF among knee OA patients, but further research is required to establish causal relationships.

## Introduction

Knee osteoarthritis (OA) is a prevalent condition affecting millions of people worldwide, particularly the elderly. In 2020, it was estimated that approximately 654.1 million individuals

**Funding:** The author(s) received no specific funding for this work.

**Competing interests:** The authors have declared that no competing interests exist

aged 40 and older had knee OA globally [1]. Knee OA is a complex and multifactorial condition involving not only the joint cartilage but also other joint structures, such as ligaments, menisci, and subchondral bone. It is also influenced by systemic factors such as inflammation, metabolic disorders, and hormonal changes [2, 3].

In addition to joint-related symptoms, knee OA is an independent risk factor of falls and fractures [3, 4], with vertebral fractures (VF) being the most common site for osteoporotic fractures (6). Approximately two-thirds of VF cases go unnoticed in a clinical setting, often referred to as asymptomatic morphometric vertebral fracture (AMVF) [5–7]. The ROAD study demonstrated a significant association between VF and knee OA with lower physical quality of life (QOL) scores in men over 40 years old. The impact of VF on physical QOL is greater than that of a cerebral stroke, highlighting the serious and potentially long-term consequences of this condition and underscoring the importance of early detection and treatment, particularly in older male populations [8].

Various factors contribute to the risk of VF in patients with knee OA. These include patient-specific factors such as age, sex, body mass index (BMI), smoking, and comorbidities, as well as disease-specific factors such as the severity and duration of joint damage, muscle weakness, and impaired balance [9–12].

The loss of skeletal muscle mass, or pre-sarcopenia, which mostly occurs with aging, can impair physical function and balance, thereby increasing the risk of falls and fractures [13, 14]. Obesity, which is common in knee OA, can increase mechanical stress on the joints and also affect bone metabolism, leading to reduced bone mineral density (BMD) and increased fracture risk [15, 16]. BMD, a measure of bone strength, is an important determinant of fracture risk, and low BMD is a well-established risk factor for VF in both men and women [17].

Despite the prevalence of knee OA and its associated increased risk of falls and fractures, there is the lack of a specific scoring system to predict fracture risk in knee OA patients. While commonly used validated scoring systems such as Knee Injury and Osteoarthritis Outcome Score, and Western Ontario and McMaster Universities Osteoarthritis Index assess joint pain, stiffness, and physical function, they do not directly evaluate fracture risk [18]. Furthermore, existing fracture risk assessment tools such as FRAX and Garvan Fracture Risk Calculator are not specific to knee OA patients and even include OA as a risk factor [19]. Overall, there is a pressing need for more specialized tools that can accurately evaluate fracture risk in knee OA patients and facilitate targeted preventive measures to reduce the risk of falls and fractures in this population.

In this study, we assessed and evaluated the relationship of potential predictive factors associated with AMVF among knee OA patients. The results of this study may aid healthcare professionals to identify knee OA patients at increased risk of fracture and to implement appropriate interventions for preventing fractures and improving patient outcomes.

## Materials and methods

### Study design and sampling procedures

This cross-sectional study was conducted at the orthopedics and rheumatology clinic at Hospital Pengajar Universiti Putra Malaysia (HPUPM). The study spanned from October 2022 to March 2023. The study population consisted of outpatient patients with specific knee osteoarthritis from the orthopedic and rheumatology department at HPUPM. A convenience sampling method was employed for this study. Potential candidates were selected from the list of patients at the orthopedics and rheumatology clinic, and only those who met the inclusion criteria were included in the study.

## Participant eligibility criteria

The study's inclusion criteria specified participants who were patients diagnosed with knee osteoarthritis using the Kellgren/Lawrence grading system (minimum grade of 2) by an orthopedic or rheumatologist, and who were aged over 50 years. The exclusion criteria were comprehensive to maintain the study's integrity. Those excluded were patients diagnosed with osteoporosis or receiving osteoporosis medication, taking drugs affecting bone homeostasis such as high-dose corticosteroids, phenytoin, methotrexate, cyclosporine, or oral contraceptive pills [20], or having a known metabolic bone disorder. Also excluded were individuals with malabsorption issues [21], thyroid disease, malignancy [22], stage 3 and above chronic kidney disease (CKD) with an eGFR less than 60 ml/min using the Cockroft-Gault formula [23], and those with a history of trauma or surgical intervention on the spine. Additionally, patients unable to give consent or lie down during a whole-body dual-energy X-ray absorptiometry (DEXA) scan and pregnant women were not included in this study.

## Data collection

Data collection for this study began with the acquisition of an initial patient list from the Orthopedics and Rheumatology departments at HPUPM, secured with the department head's approval and the HPUPM's director. Patient data were collected from both electronic patient records and questionnaires. Electronic records were used to collect data on patients' socio-demographic profiles, comorbidities, comprehensive diagnoses, and examination results, including BMI. Additional information, such as detailed smoking history, occupational details, and menopause status for women, was collected through questionnaires during patients' clinic visits. All collected data was then transcribed into the study's proforma.

Upon the patients' first visit to the clinic, the assigned physician in the orthopedic or rheumatology clinic undertook the task of assessing the patients' conditions. Any necessary modifications on the medication were based on the physician's clinical expertise and the specific needs of the individual patients. Following this, the patients were introduced to the research team, and their informed consent for participation in the study was obtained. A whole body DEXA scan, encompassing BMD and vertebral fracture assessment (VFA), was also scheduled for each patient during this initial visit. It is imperative to underscore that the research team neither added nor reduced any medication during this stage.

## Outcome and risk factor definitions

The primary outcome of this study was the presence of asymptomatic morphometric vertebral fractures (AMVF). AMVF was assessed using vertebral fracture assessment (VFA) as part of the whole-body DEXA scan. Vertebral fractures were identified and graded according to semi-quantitative methods as described previously [24].

The variables of this study are categorized into independent and dependent variables. Independent variables include sociodemographic factors such as age, gender, race, smoking status, menopause status for women, duration of knee osteoarthritis, and whether the patient underwent knee arthroplasty. Comorbidities were also considered as independent variables including diabetes mellitus, hypertension, dyslipidemia, and ischemic heart disease (IHD). Furthermore, indicators of pre-sarcopenia, derived from the Appendicular Skeletal Muscle Mass from DEXA, lumbar BMD, and obesity status, determined through BMI or fat mass from DEXA, were also included as independent variables. Fat mass percentage was derived from the whole-body DEXA scan, with the scanner software calculating total body fat mass and providing this as a percentage of total body mass. BMI was calculated by dividing the patient's weight in kilograms by the square of their height in meters ($kg/m^2$). Lumbar BMD

was measured directly by the DEXA scan, focusing on the L1-L4 vertebrae, and reported in g/cm$^2$. On the other hand, the dependent variables in this study were the occurrence, location, type, severity, and number of AVMF.

## Study ethics and patient confidentiality procedures

This study was approved by the Ethics Committee for Research involving human subjects of University Putra Malaysia (JKEUPM) on October 19, 2022, with the reference number UPM/TNCPI/RMC/JKEUPM/1.4.18.2 (JKEUPM) JKEUPM-2022-523. Prior to data collection, the necessary approvals were also secured from the orthopedic and rheumatology clinic involved. This study was strictly conducted in accordance with ethical principles, including the protection of subject vulnerability, absence of conflict of interest, privacy and confidentiality, sensitivity to community considerations, and ensuring benefits to the participants. To ensure patient confidentiality, a secure, password-protected database was used to store all names, linked only to a study identification number. This identification number replaced patient identifiers on subject data sheets. Data entry was completed on a password-protected computer. Both hard and soft copies of personal data, including medical records and study data, will be archived for a period of five years. After this period, all study data and documents will be properly disposed of, destroyed, or deleted in accordance with established protocols. In terms of publication policy, no personal information will be disclosed and subjects will not be identifiable in any published survey findings. The de-identified raw data of this study have been deposited to Zenodo: https://zenodo.org/records/12792011 and this record will be deleted after October 18, 2027, in accordance with the approved human ethics for this study.

## Statistical analysis

The normality of variable distribution was assessed using Shapiro-Wilk test. The chi-squared test or Fisher's exact test were utilized as appropriate for comparing categorical variables across multiple groups. For comparison between two groups of continuous variables, data with normal or non-normal distribution were tested using independent t-test or Mann-Whitney test, respectively. Multivariable logistic regression analysis was conducted by including multiple variables simultaneously to determine their individual associations with AMVF in knee OA patients after controlling for the effects of other variables. Our multivariable logistic regression model included variables that demonstrated significance in the univariable analysis, as well as variables that showed a trend towards significance. We also included other comorbidities (dyslipidemia, hypertension, IHD) to account for patients presenting with multiple comorbidities. All selected variables were included together in the multivariable analysis. A $p$-value below 0.05 was used to determine statistical significance. All statistical analyses were conducted using IBM SPSS Statistics (version 29.0).

## Results

### Patient's demographic and clinical characteristics

Our study included a total of 76 patients diagnosed with knee OA. The demographic characteristics of the patients are presented in Table 1. The mean age of the patients was 66.39 ± 5.24 years, with 61.8% of them being 65 years or older. The majority of the patients were female (84.2%) and of Malay ethnicity (84.2%). Most of the patients were non-smokers (98.7%), with a mean height of 155.61±7.79 cm and a mean weight of 74.04±10.48 kg. The mean BMI was 30.65±4.83 kg/m$^2$, with 55.3% of the patients being classified as obese.

**Table 1. Clinico-demographic characteristics of knee OA patients (n = 76).**

| Variables | Mean ± SD or n (%) |
|---|---|
| **Age (years)** | 66.39 ± 5.24 |
| <65 | 29 (38.2) |
| ≥65 | 47 (61.8) |
| **Gender** | |
| Male | 12 (15.8) |
| Female | 64 (84.2) |
| **Ethnicity** | |
| Malay | 64 (84.2) |
| Chinese | 3 (3.9) |
| Indian | 9 (11.8) |
| **Smoking** | |
| No | 75 (98.7) |
| Yes | 1 (1.3) |
| **Measurements** | |
| Height (cm) | 155.61 ± 7.79 |
| Weight (kg) | 74.04 ± 10.48 |
| **BMI (kg/m$^2$)** | 30.65 ± 4.83 |
| Normal | 8 (10.5) |
| Overweight | 26 (34.2) |
| Obese | 42 (55.3) |
| **Menopause status** | |
| No | 1 (1.6) |
| Yes | 63 (98.4) |
| **Occupation** | |
| Sedentary | 3 (3.9) |
| Light work | 25 (32.9) |
| Moderate work | 35 (46.1) |
| Heavy work | 13 (17.1) |
| **Comorbidities*** | |
| Diabetes mellitus | 37 (48.7) |
| Hypertension | 52 (68.4) |
| Dyslipidemia | 55 (72.4) |
| IHD | 6 (7.9) |
| **Duration of knee OA** | |
| <10 years | 43 (56.6) |
| ≥10 years | 33 (43.4) |
| **Knee arthroplasty** | |
| No | 57 (75.0) |
| Yes | 19 (25.0) |
| **Pre-sarcopenia status** | |
| Normal | 60 (78.9) |
| Pre-sarcopenia | 16 (21.1) |
| **Lumbar BMD category** | |
| Normal | 31 (40.8) |
| Osteopenia | 36 (47.4) |
| Osteoporosis | 9 (11.8) |

*Certain patients presented with multiple comorbidities.

In terms of comorbidities, 48.7% of the patients had diabetes mellitus, 68.4% had hypertension, 72.4% had dyslipidemia, and 7.9% had IHD. The duration of knee OA was less than 10 years for 56.6% of the patients and 10 years or more for 43.4% of the patients. A quarter of the patients (25.0%) had undergone knee arthroplasty. Pre-sarcopenia was observed in 21.1% of the patients. The lumbar BMD was categorized as normal in 40.8% of the patients, osteopenia in 47.4%, and osteoporosis in 11.8%.

### Association of categorical and continuous variables with AMVF in knee OA patients

The data was analyzed to determine the association of categorical variables with the presence of AMVF in knee OA patients (Table 2). The variables considered in this analysis included occupation, diabetes mellitus, hypertension, dyslipidemia, and IHD. The analysis revealed a significant association between occupation and AMVF. Specifically, patients engaged in moderate or heavy work had a higher incidence of AMVF ($p<0.001$). In addition to occupation, the presence of diabetes mellitus also showed a significant association with AMVF ($p = 0.016$). However, other comorbidities such as hypertension, dyslipidemia, and IHD did not show a significant association with AMVF, indicating that these conditions did not have a significant association with the presence of AMVF in knee OA patients.

Our study also examined the association of continuous variables with AMVF in knee OA patients (Fig 1). The continuous variables considered in this analysis included fat mass percentage, BMI, lumbar BMD, age, height, and weight. The analysis demonstrated a significant association between fat mass percentage and AMVF ($p = 0.027$). BMI and lumbar BMD showed a non-significant trend towards AMVF, with $p$-values of 0.053 and 0.096, respectively. Other variables including age, height, and weight did not show a significant association with AMVF in knee OA patients.

### Multivariable logistic regression analysis on the variables associated with AMVF in knee OA patients

A multivariable logistic regression analysis was performed to identify the variables associated with AMVF in knee OA patients. The variables included those that demonstrated significance in the univariable analysis (*i.e.* occupation, diabetes mellitus comorbidity, fat mass), and BMI and lumbar BMD due to both of these variables showed a trend towards significance in univariable analysis. Other comorbidities (*i.e.* dyslipidemia, hypertension, IHD) were also included in the multivariable model to account for patients presented with multiple comorbidities. The analysis showed that occupation, specifically moderate or heavy work, demonstrated a significant association with AMVF [Hazard ratio (HR): 57.76, 95% confidence interval (CI): 4.23–788.57; $p = 0.002$] (Table 3). Furthermore, an increase in fat mass percentage was significantly associated with a decrease in AMVF occurrence in knee OA patients (HR: 0.83, 95% CI: 0.70–0.97; $p = 0.018$). BMI (HR: 1.24, 95% CI: 1.00–1.54; $p = 0.053$) and lumbar BMD (HR: 0.56, 95% CI: 0.31–1.01; $p = 0.054$) showed trends towards significance. Other factors including comorbidities were not significantly associated with AMVF: dyslipidemia (HR: 0.21, 95% CI: 0.04–1.28; $p = 0.091$), diabetes mellitus (HR: 3.08, 95% CI: 0.62–15.40; $p = 0.170$), hypertension (HR: 2.56, 95% CI: 0.42–15.41; $p = 0.306$), and IHD (HR: 0.60, 95% CI: 0.05–7.40; $p = 0.690$) (Table 3).

### Discussion

In this study, further insights were provided pertaining to the potential factors associated with AMVF in knee OA patients. We identified a significant association of AMVF with moderate or heavy work occupation, and a significant inverse association with fat mass percentage.

**Table 2. Association of categorical variables with AMVF status of knee OA patients (n = 76).**

| Variables | Without AMVF (n = 53); Mean ± SD or n (%) | With AMVF (n = 23); Mean ± SD or n (%) | p-value |
|---|---|---|---|
| **Age** | | | |
| <65 years old | 22 (28.9) | 7 (9.2) | 0.361 |
| ≥65 years old | 31 (40.8) | 16 (21.1) | |
| **Gender** | | | |
| Male | 6 (7.9) | 6 (7.9) | 0.105 |
| Female | 47 (61.8) | 17 (22.4) | |
| **Ethnicity** | | | |
| Malay | 45 (61.8) | 19 (25.0) | >0.999 |
| Chinese | 2 (2.6) | 1 (1.3) | |
| Indian | 6 (7.9) | 3 (3.9) | |
| **Smoking** | | | |
| No | 53 (69.7) | 22 (28.9) | 0.303 |
| Yes | 0 (0) | 1 (1.3) | |
| **BMI categories** | | | |
| Normal | 4 (5.3) | 4 (5.3) | 0.288 |
| Overweight | 17 (22.4) | 9 (11.8) | |
| Obese | 32 (42.1) | 10 (13.2) | |
| **Menopause** | | | |
| No | 1 (1.6) | 0 (0) | >0.999 |
| Yes | 46 (71.9) | 17 (26.6) | |
| **Occupation** | | | |
| Sedentary or light work | 27 (35.5) | 1 (1.3) | **<0.001** |
| Moderate or heavy work | 26 (34.2) | 22 (28.9) | |
| **Comorbidities** | | | |
| Diabetes mellitus | 21 (27.6) | 16 (21.1) | **0.016** |
| No diabetes mellitus | 32 (42.1) | 7 (9.2) | |
| Hypertension | 34 (44.7) | 18 (23.7) | 0.224 |
| No hypertension | 19 (25.0) | 5 (6.6) | |
| Dyslipidemia | 39 (51.3) | 16 (21.1) | 0.718 |
| No dyslipidemia | 14 (18.4) | 7 (9.2) | |
| IHD | 4 (5.3) | 2 (2.6) | >0.999 |
| No IHD | 49 (64.5) | 21(27.6) | |
| **Duration of Knee OA** | | | |
| ≥10 years | 30 (39.5) | 13 (17.1) | >0.999 |
| <10 years | 23(30.3) | 10 (13.2) | |
| **Knee Arthroplasty** | | | |
| Yes | 15 (19.7) | 4 (5.3) | 0.313 |
| No | 38 (50.0) | 19 (25.0) | |
| **Pre-sarcopenia status** | | | |
| Normal | 43 (56.6) | 17 (22.4) | 0.545 |
| Pre-sarcopenia | 10 (13.2) | 6 (7.9) | |
| **Lumbar BMD status** | | | |
| Normal | 25 (32.9) | 6 (7.9) | 0.217 |
| Osteopenia | 22 (28.9) | 14 (18.4) | |
| Osteoporosis | 6 (7.9) | 3 (3.9) | |

Significant p-value is in bold.

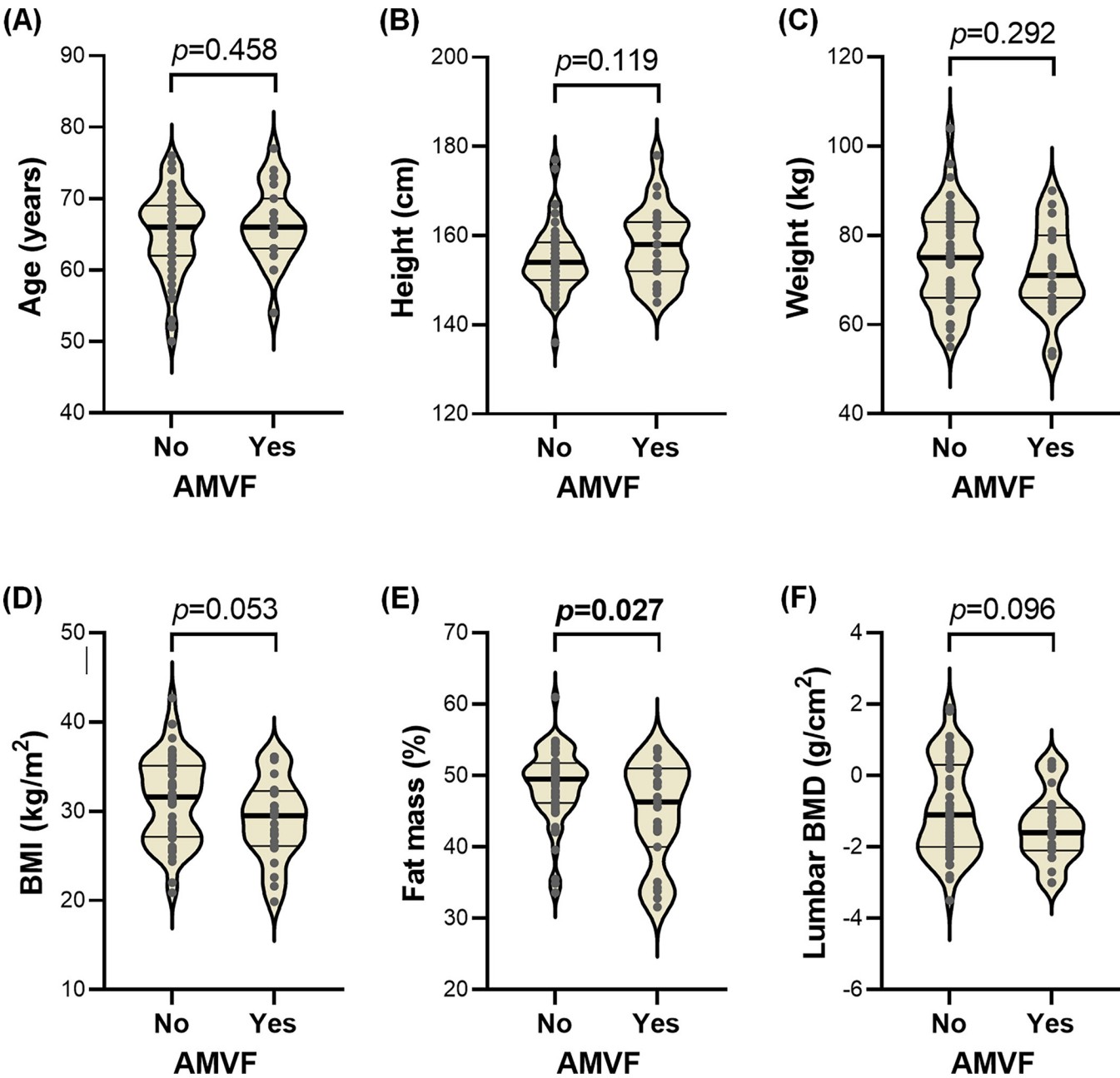

**Fig 1. Association of knee OA AMVF patients with continuous variables.** (A) Age (years); (B) Height (cm); (C) Weight (kg); (D) BMI (kg/m2); (E) Fat mass (%); (F) Lumbar BMD (g/cm2). Within each violin plot, the upper, middle (darker), and lower lines denote the third quartile, median, and first quartile, respectively.

In our cohort of knee OA patients, those engaged in moderate or heavy work demonstrated a higher incidence of AMVF. This is consistent with multiple past studies which have demonstrated that physically demanding occupations or heavy manual labor can increase the risk of musculoskeletal disorders including knee OA [25, 26]. Notably, a longitudinal, multiple-cohort study showed that individuals with heavy manual occupations demonstrated a two-fold increased risk of radiographic knee OA [26]. In a separate international participant-level cohort study, a two-fold increase risk of having radiographic knee OA, knee pain, and

**Table 3. Multivariable logistic regression analysis on the variables associated with AMVF in knee OA patients (n = 76).**

| Variables | B coefficient | HR (95% confidence interval) | p-value |
|---|---|---|---|
| Occupation (Moderate or heavy work) | 4.06 | 57.76 (4.23–788.57) | **0.002** |
| Fat mass (Increasing percentage) | -0.19 | 0.83 (0.70–0.97) | **0.018** |
| BMI (Increasing kg/m$^2$) | 0.22 | 1.24 (1.00–1.54) | 0.053 |
| Lumbar BMD (Increasing g/cm$^2$) | -0.58 | 0.56 (0.31–1.01) | 0.054 |
| Comorbidity (Dyslipidemia) | -1.55 | 0.21 (0.04–1.28) | 0.091 |
| Comorbidity (Diabetes mellitus) | 1.13 | 3.08 (0.62–15.40) | 0.170 |
| Comorbidity (Hypertension) | 0.94 | 2.56 (0.42–15.41) | 0.306 |
| Comorbidity (IHD) | -0.51 | 0.60 (0.05–7.40) | 0.690 |

Significant p-value is in bold.

symptomatic radiographic knee OA was also observed in those with heavy manual occupations compared with sedentary occupations [25]. Likewise, our multivariable analysis indicates that physically demanding occupations may increase the risk of developing AMVF in knee OA patients. Further research is recommended to determine the mechanisms governing this association.

Interestingly, we observed that an increase in fat mass percentage, but not BMI, was associated with a decrease in AMVF occurrence in knee OA patients. This is comparable with the findings of another cross-sectional study of Malaysian knee OA patients whereby a low body fat percentage was inversely associated with AMVF prevalence in univariable analysis [7]. However, these observations are partially inconsistent with recent studies that demonstrated higher waist circumference was associated with increased risk of fracture in men [27]. Nonetheless, this study also found that BMI was not significantly associated with fracture risk, similar with another study which demonstrated that AMVF was not significantly associated with BMI categories [7]. Likewise, another study also concluded that BMI was not significantly associated with bone fracture [28]. These studies align with our findings that BMI was not significantly associated with fracture risk.

To the best of our knowledge, a direct mechanistic link between an increase in fat mass percentage and a decrease in AMVF occurrence in knee OA patients has not been demonstrated. It is noteworthy that our study examined fat mass percentage, which may provide a more precise measure of body composition compared with BMI or waist circumference. Findings of our study suggest that a higher body fat percentage may offer some protective benefits against AMVF in knee OA patients, potentially due to the capacity of fat tissues to absorb physical pressure. However, this remains unclear and warrants further investigation.

We acknowledge the limitations of our study as follows: 1) The cross-sectional design of this study does not allow for the determination of causal relationships between the factors examined and AMVF; 2) The small sample size of 76 patients confers limited statistical power, and our results should thus be considered exploratory and descriptive rather than definitive; 3) Our study showed an association between higher fat mass percentage and lower AMVF occurrence, but this finding should be interpreted cautiously. The cross-sectional design and small sample size limit causal inferences. Hence, the complex relationship between body composition and bone health requires further investigation.

Nevertheless, while DEXA scans are commonly used to assess BMD, they do not comprehensively evaluate the factors associated with fractures in knee OA patients. Therefore, it has been recommended to include a whole-body DEXA scan to assess sarcopenia and fat mass, as well as a VFA to detect vertebral fractures in addition to BMD by DEXA, in order to provide a

more comprehensive assessment of fracture risk [29–31]. In this study, we utilized whole-body DEXA to assess BMD, muscle mass, and fat mass, providing a more extensive understanding of the relationship between these factors and vertebral fractures in this population. Furthermore, we used VFA instead of traditional X-rays to evaluate VF in order to improve sensitivity and specificity of the detection [31].

Overall, this study provides further insights into the factors associated with AMVF in knee OA patients, supporting the potential importance of considering patients' occupational activities in the management and prevention of AMVF in knee OA patients. However, larger longitudinal studies are required to establish causal relationships and confirm these findings. Finally, our exploratory study found an unexpected association between higher fat mass percentage and lower AMVF occurrence in knee OA patients. This highlights the complex relationship between body composition and bone health, underscoring the need for longitudinal studies.

## Author Contributions

**Conceptualization:** Hakimah Mohammad Sallehudin, Fahrudin Che Hamzah, Norafida Bahari.

**Data curation:** Izzatul Nadiah Zolkiply, Kah Keng Wong, Hakimah Mohammad Sallehudin, Mohammad Zulkarnain Bidin.

**Formal analysis:** Kah Keng Wong, Mohammad Zulkarnain Bidin, Norafida Bahari.

**Investigation:** Izzatul Nadiah Zolkiply, Norafida Bahari.

**Methodology:** Izzatul Nadiah Zolkiply, Hakimah Mohammad Sallehudin, Mohammad Zulkarnain Bidin.

**Project administration:** Izzatul Nadiah Zolkiply.

**Resources:** Izzatul Nadiah Zolkiply, Kah Keng Wong, Mohammad Zulkarnain Bidin.

**Software:** Kah Keng Wong, Mohammad Zulkarnain Bidin.

**Supervision:** Hakimah Mohammad Sallehudin, Fahrudin Che Hamzah, Norafida Bahari, Wan Syamimee Wan Ghazali.

**Validation:** Kah Keng Wong, Fahrudin Che Hamzah, Norafida Bahari, Wan Syamimee Wan Ghazali.

**Visualization:** Kah Keng Wong, Hakimah Mohammad Sallehudin, Wan Syamimee Wan Ghazali.

**Writing – original draft:** Izzatul Nadiah Zolkiply.

**Writing – review & editing:** Kah Keng Wong, Hakimah Mohammad Sallehudin, Wan Syamimee Wan Ghazali.

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
