## [Decision Letter · Decision Letter 0]

4 Jul 2024

PONE-D-23-20551Work Intensity and Fat Mass Percentage are Risk Factors for Asymptomatic Morphologic Vertebral Fractures in Knee Osteoarthritis PatientsPLOS ONE

Dear Dr. Wan Ghazali,

Thank you for submitting your manuscript to PLOS ONE. After careful consideration, we feel that it has merit but does not fully meet PLOS ONE’s publication criteria as it currently stands. Therefore, we invite you to submit a revised version of the manuscript that addresses the points raised during the review process.

We look forward to receiving your revised manuscript.

Kind regards,

Zhifeng Yu

Academic Editor

PLOS ONE

Journal Requirements:

3. We note you have included a table to which you do not refer in the text of your manuscript. Please ensure that you refer to Table 3 in your text; if accepted, production will need this reference to link the reader to the Table.

Additional Editor Comments:

The paper reads well and seeks to answer an important question of risk factors for asymptomatic morphological vertebral fractures in OA. Please see my comments below.

Please indicate the study design in the study title in accordance with study reporting guidelines e.g. STROBE

Report the prevalence of OA in this sentence instead of saying ‘millions’: “Knee osteoarthritis (OA) is a common condition affecting millions of individuals globally”.

A cross sectional study is not a suitable study design to assess risk factors for outcomes of interest (i.e., to derive causal relationships). Therefore, the authors of this work should refrain from claiming definitive conclusion regarding the association of work intensity and fat mass percentage on AMVF. Additionally, the sample size of 76 patients is very small for risk factor analysis in observational studies. Ideally, this should be a descriptive study.

If fat mass percentage associated with a decreased diagnosis of AMVF, how is it a risk factor?

Please tone down the postulated implication of this study. First, as I already mentioned, you cannot make inference from a cross-sectional study. Noted also that risk factor identification is a step towards producing a risk assessment tool which aids the identification of at-risk population.

Did you access electronic patient records or were data collected by questionnaires?

Please provide a sub-section on how the outcome was defined separate from that of how risk factors were defined., e.g. how was fat mass percentage derived? And was this already recorded in the data used or was it calculated by the research team?

Could you elaborate on the model building strategy you employed? You mention including variables in the model simultaneously, but it is unclear what this means.

Please be consistent in reporting OR (95% CI) in addition to P values in your results. Avoid reporting only P-values.

Could the authors not downplay the fact that sample size for this study was small while stating this as a limitation? I am surprised that this is the only limitation acknowledged.

Reviewers' comments:

Reviewer's Responses to Questions

**Comments to the Author**

1. Is the manuscript technically sound, and do the data support the conclusions?

Reviewer #1: Yes

Reviewer #2: No

2. Has the statistical analysis been performed appropriately and rigorously? 

Reviewer #1: Yes

Reviewer #2: Yes

3. Have the authors made all data underlying the findings in their manuscript fully available?

Reviewer #1: Yes

Reviewer #2: No

4. Is the manuscript presented in an intelligible fashion and written in standard English?

Reviewer #1: Yes

Reviewer #2: Yes

5. Review Comments to the Author

Reviewer #1: The document presents scientific relevance and the authors objectively describe what was proposed, and the results are clearly exposed. The discussion is well directed to the results found, as well as the conclusion.

Reviewer #2: Thank you for the opportunity to review Ghazali et al manuscript on “Work Intensity and Fat Mass Percentage are Risk Factors for Asymptomatic Morphologic Vertebral Fractures in Knee Osteoarthritis Patients”. The paper reads well and seeks to answer an important question of risk factors for asymptomatic morphological vertebral fractures in OA. Please see my comments below.

Please indicate the study design in the study title in accordance with study reporting guidelines e.g. STROBE

Report the prevalence of OA in this sentence instead of saying ‘millions’: “Knee osteoarthritis (OA) is a common condition affecting millions of individuals globally”.

A cross sectional study is not a suitable study design to assess risk factors for outcomes of interest (i.e., to derive causal relationships). Therefore, the authors of this work should refrain from claiming definitive conclusion regarding the association of work intensity and fat mass percentage on AMVF. Additionally, the sample size of 76 patients is very small for risk factor analysis in observational studies. Ideally, this should be a descriptive study.

If fat mass percentage associated with a decreased diagnosis of AMVF, how is it a risk factor?

Please tone down the postulated implication of this study. First, as I already mentioned, you cannot make inference from a cross-sectional study. Noted also that risk factor identification is a step towards producing a risk assessment tool which aids the identification of at-risk population.

Did you access electronic patient records or were data collected by questionnaires?

Please provide a sub-section on how the outcome was defined separate from that of how risk factors were defined., e.g. how was fat mass percentage derived? And was this already recorded in the data used or was it calculated by the research team?

Could you elaborate on the model building strategy you employed? You mention including variables in the model simultaneously, but it is unclear what this means.

Please be consistent in reporting OR (95% CI) in addition to P values in your results. Avoid reporting only P-values.

Could the authors not downplay the fact that sample size for this study was small while stating this as a limitation? I am surprised that this is the only limitation acknowledged.

6. PLOS authors have the option to publish the peer review history of their article (what does this mean?). If published, this will include your full peer review and any attached files.

Reviewer #1: No

Reviewer #2: No

---

## [Author Response · Author response to Decision Letter 0]

28 Jul 2024

Journal Requirements:

Thank you for the comments. We have ensured that the manuscript meets PLOS ONE’s style requirements accordingly

Thank you for the comments. We have deposited the raw data required to reproduce our study’s results including the clinical, demographical (without any personal identifiers and all patients have been anonymized) and other relevant characteristics, as well as the data and graphs for Figure 1, to Zenodo: https://zenodo.org/records/12792011

As stated in our manuscript: “This study was approved by the Ethics Committee for Research involving human subjects of University Putra Malaysia (JKEUPM) on October 19, 2022, with the reference number UPM/TNCPI/RMC/JKEUPM/1.4.18.2 (JKEUPM) JKEUPM-2022-523. Prior to data collection, the necessary approvals were also secured from the orthopedic and rheumatology clinic involved. This study was strictly conducted in accordance with ethical principles, including the protection of subject vulnerability, absence of conflict of interest, privacy and confidentiality, sensitivity to community considerations, and ensuring benefits to the participants. 

To ensure patient confidentiality, a secure, password-protected database was used to store all names, linked only to a study identification number. This identification number replaced patient identifiers on subject data sheets. Data entry was completed on a password-protected computer. Both hard and soft copies of personal data, including medical records and study data, will be archived for a period of five years. After this period, all study data and documents will be properly disposed of, destroyed, or deleted in accordance with established protocols. In terms of publication policy, no personal information will be disclosed and subjects will not be identifiable in any published survey findings.” 

Therefore, the raw data deposited to Zenodo will be deleted after October 18, 2027, in accordance with the approved human ethics for this study. We have added these new descriptions in the Study Ethics and Patient Confidentiality Procedures section: 

The de-identified raw data of this study have been deposited to Zenodo: https://zenodo.org/records/12792011 and this record will be deleted after October 18, 2027, in accordance with the approved human ethics for this study.

3. We note you have included a table to which you do not refer in the text of your manuscript. Please ensure that you refer to Table 3 in your text; if accepted, production will need this reference to link the reader to the Table.

Thank you for the comments. Table 3 has now been referred in the text as follows (Results section): 

The analysis showed that occupation, specifically moderate or heavy work, demonstrated a significant association with AMVF [Hazard ratio (HR): 57.76, 95% confidence interval (CI): 4.23-788.57; p=0.002] (Table 3).

Thank you for the comments. The reference list has been reviewed accordingly.

Additional Editor Comments:

The paper reads well and seeks to answer an important question of risk factors for asymptomatic morphological vertebral fractures in OA. Please see my comments below.

Please indicate the study design in the study title in accordance with study reporting guidelines e.g. STROBE

Thank you very much for the comments. The title has now been changed to: 

Work Intensity and Fat Mass Percentage are Associated with Asymptomatic Morphometric Vertebral Fractures in Knee Osteoarthritis Patients: A Cross-Sectional Study

Report the prevalence of OA in this sentence instead of saying ‘millions’: “Knee osteoarthritis (OA) is a common condition affecting millions of individuals globally”.

Thank you for the comments. The sentence has now been revised as follows:

Knee osteoarthritis (OA) is a common condition with a prevalence of 365 million individuals globally

A cross sectional study is not a suitable study design to assess risk factors for outcomes of interest (i.e., to derive causal relationships). Therefore, the authors of this work should refrain from claiming definitive conclusion regarding the association of work intensity and fat mass percentage on AMVF. Additionally, the sample size of 76 patients is very small for risk factor analysis in observational studies. Ideally, this should be a descriptive study.

Thank you for the comments. We have now revised the required parts to tone down previous statements of the manuscript as follows:

Abstract section: "These findings support the potential importance of considering occupational activities and body fat composition in managing AMVF among knee OA patients, but further research is required to establish causal relationships"

Revised limitations of the Discussion section: "1) The cross-sectional design of this study does not allow for the determination of causal relationships between the factors examined and AMVF; 2) The small sample size of 76 patients confers limited statistical power, and our results should thus be considered exploratory and descriptive rather than definitive."

Conclusion (the final paragraph of the manuscript): “Overall, this study provides further insights into the factors associated with AMVF in knee OA patients, supporting the potential importance of considering patients’ occupational activities in the management and prevention of AMVF in knee OA patients. However, larger longitudinal studies are required to establish causal relationships and confirm these findings.”

If fat mass percentage associated with a decreased diagnosis of AMVF, how is it a risk factor?

Thank you for the comments. In the conclusion paragraph, we have removed the phrase “risk factors” and to simply state “factors” for the sentence “Overall, this study provides further insights into the factors associated with AMVF in knee OA patients”. 

We have also added the following new limitation of the study in the Discussion section: 

3) Our study showed an association between higher fat mass percentage and lower AMVF occurrence, but this finding should be interpreted cautiously. The cross-sectional design and small sample size limit causal inferences. Hence, the complex relationship between body composition and bone health requires further investigation.

The conclusion paragraph (the final paragraph of the manuscript) has also been revised as follows: 

Finally, our exploratory study found an unexpected association between higher fat mass percentage and lower AMVF occurrence in knee OA patients. This highlights the complex relationship between body composition and bone health, underscoring the need for longitudinal studies.

Please tone down the postulated implication of this study. First, as I already mentioned, you cannot make inference from a cross-sectional study. Noted also that risk factor identification is a step towards producing a risk assessment tool which aids the identification of at-risk population.

Thank you for the comments.

The conclusion paragraph has been revised as follows:

Previous version: Overall, this study provides further insights into the risk factors associated with AMVF in knee OA patients, highlighting the importance of considering patients’ occupational activities in the management and prevention of AMVF in knee OA patients. Moreover, our findings demonstrate the potential role of fat body composition in fracture risk, suggesting that this may be an important area of future research in knee OA patients.

Revised version: Overall, this study provides further insights into the factors associated with AMVF in knee OA patients, supporting the potential importance of considering patients’ occupational activities in the management and prevention of AMVF in knee OA patients. However, larger longitudinal studies are required to establish causal relationships and confirm these findings. Finally, our exploratory study found an unexpected association between higher fat mass percentage and lower AMVF occurrence in knee OA patients. This highlights the complex relationship between body composition and bone health, underscoring the need for longitudinal studies.

In addition, when describing our results, the term "risk factors" have been removed from the title, discussion, and conclusion. 

Did you access electronic patient records or were data collected by questionnaires?

Thank you for the comments. Patient’s data were collected from both electronic patients records as well as from questionnaires, and the follow descriptions have been added in the Materials and Methods section: 

Patient data were collected from both electronic patient records and questionnaires. Electronic records were used to collect data on patients’ socio-demographic profiles, comorbidities, comprehensive diagnoses, and examination results, including BMI. Additional information, such as detailed smoking history, occupational details, and menopause status for women, was collected through questionnaires during patients’ clinic visits. All collected data was then transcribed into the study’s proforma.

Please provide a sub-section on how the outcome was defined separate from that of how risk factors were defined., e.g. how was fat mass percentage derived? And was this already recorded in the data used or was it calculated by the research team?

Thank you for the comments. The following descriptions have been added in the Materials and Methods section: 

Outcome and Risk Factor Definitions

The primary outcome of this study was the presence of asymptomatic morphometric vertebral fractures (AMVF). AMVF was assessed using vertebral fracture assessment (VFA) as part of the whole-body DEXA scan. Vertebral fractures were identified and graded according to semi-quantitative methods as described previously [24].

Fat mass percentage was derived from the whole-body DEXA scan, with the scanner software calculating total body fat mass and providing this as a percentage of total body mass. BMI was calculated by dividing the patient's weight in kilograms by the square of their height in meters (kg/m²). Lumbar BMD was measured directly by the DEXA scan, focusing on the L1-L4 vertebrae, and reported in g/cm².

Could you elaborate on the model building strategy you employed? You mention including variables in the model simultaneously, but it is unclear what this means.

Thank you for the comments. The following descriptions have been added in the Statistical Analysis section: 

Our multivariable logistic regression model included variables that demonstrated significance in the univariable analysis (occupation, diabetes mellitus comorbidity, fat mass), as well as variables that showed a trend towards significance (BMI and lumbar BMD). We also included other comorbidities (dyslipidemia, hypertension, IHD) to account for patients presenting with multiple comorbidities. All selected variables were included together in the multivariable analysis.

Please be consistent in reporting OR (95% CI) in addition to P values in your results. Avoid reporting only P-values.

Thank you for the comments. The following has now been added in the Results section: 

BMI (HR: 1.24, 95% CI: 1.00-1.54; p=0.053) and lumbar BMD (HR: 0.56, 95% CI: 0.31-1.01; p=0.054) showed trends towards significance. Other factors including comorbidities were not significantly associated with AMVF: dyslipidemia (HR: 0.21, 95% CI: 0.04-1.28; p=0.091), diabetes mellitus (HR: 3.08, 95% CI: 0.62-15.40; p=0.170), hypertension (HR: 2.56, 95% CI: 0.42-15.41; p=0.306), and IHD (HR: 0.60, 95% CI: 0.05-7.40; p=0.690) (Table 3).

Could the authors not downplay the fact that sample size for this study was small while stating this as a limitation? I am surprised that this is the only limitation acknowledged.

Thank you for the comments. The limitations have been revised and expanded as follows: 

We acknowledge the limitations of our study as follows: 1) The cross-sectional design of this study does not allow for the determination of causal relationships between the factors examined and AMVF; 2) The small sample size of 76 patients confers limited statistical power, and our results should thus be considered exploratory and descriptive rather than definitive; 3) Our study showed an association between higher fat mass percentage and lower AMVF occurrence, but this finding should be interpreted cautiously. The cross-sectional design and small sample size limit causal inferences. Hence, the complex relationship between body composition and bone health requires further investigation.

Thank you.

---

## [Editor Report · Decision Letter 1]

30 Jul 2024

Work Intensity and Fat Mass Percentage are Associated with Asymptomatic Morphometric Vertebral Fractures in Knee Osteoarthritis Patients: A Cross-Sectional Study

PONE-D-23-20551R1

Dear Dr. Wan Ghazali,

We’re pleased to inform you that your manuscript has been judged scientifically suitable for publication and will be formally accepted for publication once it meets all outstanding technical requirements.

Kind regards,

Zhifeng Yu

Academic Editor

PLOS ONE

---

## [Editor Report · Acceptance letter]

2 Aug 2024

PONE-D-23-20551R1 

PLOS ONE

Dear Dr. Wan Ghazali, 

I'm pleased to inform you that your manuscript has been deemed suitable for publication in PLOS ONE. Congratulations! Your manuscript is now being handed over to our production team.

Kind regards, 

on behalf of

Dr. Zhifeng Yu 

Academic Editor

PLOS ONE